# NeuManifold: Neural Watertight Manifold Reconstruction with Efficient and High-Quality Rendering Support

## Abstract

We present a method for generating high-quality watertight manifold meshes from multi-view input images. Existing volumetric rendering methods are robust in optimization but tend to generate noisy meshes with poor topology. Differentiable rasterization-based methods can generate high-quality meshes but are sensitive to initialization. Our method combines the benefits of both worlds; we take the geometry initialization obtained from neural volumetric fields, and further optimize the geometry as well as a compact neural texture representation with differentiable rasterizers. Through extensive experiments, we demonstrate that our method can generate accurate mesh reconstructions with faithful appearance that are comparable to previous volume rendering methods while being an order of magnitude faster in rendering. We also show that our generated mesh and neural texture reconstruction is compatible with existing graphics pipelines and enables downstream 3D applications such as simulation.

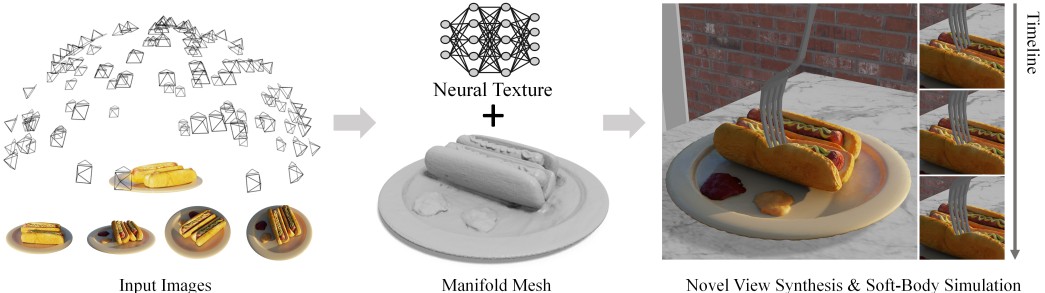

Fig. 1. NeuManifold takes 2D images as input and generates watertight manifold meshes with neural textures. NeuManifold enables many downstream applications including high-quality novel-view synthesis and soft-body simulation.

## 1 Introduction

Recent advancements in neural field representations (Mildenhall et al., 2021; Müller et al., 2022; Chen et al., 2023b) have enabled scene reconstructions with photorealistic rendering quality. However, they use volumetric representations, resulting in slow rendering and limited support for standard 3D pipelines like appearance editing, physical simulation, and geometry processing.

For many such applications, meshes—especially those that are manifold and watertight—are the preferred option. Meshes can be rendered efficiently with standard 3D rendering engines and the watertight manifold property is often favorable in many geometry processing algorithms, such as mesh boolean operations, approximate convex decomposition (Wei et al., 2022), tetrahedralization (Hang, 2015) for simulation, and volumetric point sampling to initialize particle simulation.

Although mesh reconstruction has been extensively studied in prior arts Schönberger et al. (2016); Snavely et al. (2006); Furukawa & Ponce (2010), reconstructing a high-quality mesh with realistic rendering remains a highly challenging task. To address this, recent advancements in inverse graphics through differentiable surface rendering have shown great promise, such as nvdiffrec (Munkberg et al., 2021) and nerf2mesh (Tang et al., 2022). Nevertheless, the rendering quality of these methods

still lags behind that of neural field-based methods, and their meshes are only optimized for rendering applications, resulting in non-manifold models with self-intersections that are unsuitable for simulation and other geometry processing applications.

Our objective is to bridge this gap by reconstructing high-quality meshes, that facilitate fast rendering and are broadly supported for 3D applications beyond rendering, while preserving the superior visual quality of volumetric approaches. To accomplish this, we introduce NeuManifold, a novel neural approach that can produce a high-quality, *watertight manifold* mesh of a 3D scene with neural textures. As depicted in Fig. 1, our technique achieves photo-realistic rendering quality. More significantly, our mesh-based model can be employed directly in physical simulation engines that frequently require watertight and even manifold meshes.

We achieve this by integrating advanced neural field rendering with differentiable rasterization-based mesh reconstruction techniques. We observe that volumetric neural field rendering and differentiable rasterization have mutually complementary benefits. While neural field-based approaches like TensoRF (Chen et al., 2022a) can produce high visual quality and generate density fields as scene geometry, the exported meshes, when rendered using surface rendering (rasterization), cannot retain the original high visual quality achieved with volume rendering. In contrast, differentiable mesh rasterization techniques such as nvdiffrec (Munkberg et al., 2021) directly optimize the final mesh output using rendering supervision. Yet, they are sensitive to geometry initialization and can get stuck in local minima, especially when reconstructing high-resolution meshes. (see Fig. 5). Therefore, we propose leveraging neural field reconstruction techniques to create high-quality initializations for differentiable rasterization, significantly enhancing the final mesh reconstruction quality. Additionally, we have observed that the non-linearity of the density field can lead to undesirable artifacts when using previously prevalent differentiable marching algorithms (Shen et al., 2021), as illustrated in Fig. 3. To address this issue, we introduce Differentiable Marching Cubes (DiffMC), which effectively eliminates these artifacts and results in significantly smoother surfaces.

Furthermore, we enhance the visual quality of our model by modeling appearance using neural textures instead of the traditional BRDF textures utilized in most inverse rendering methods (Munkberg et al., 2021; Luan et al., 2021). Specifically, we use TensoRF(Chen et al., 2022a) to compactly factorize a 3D neural field into axis-aligned orthogonal 2D and 1D neural textures. While TensoRF uses volume rendering, we extract features from these neural textures at surface points on a mesh and decode view-dependent colors for surface rendering in differentiable rasterization. We demonstrate that our factorized neural textures produce superior rendering quality compared to other texture representations, including iNGP-base hash grid and MLPs (see Table 4).

Our work offers the following key contributions:

- We propose NeuManifold, which excels at producing high-quality watertight manifold meshes. These meshes support not only realistic rendering but also applications in a diverse array of physical simulations and geometry processing tasks.

- We introduce the first complete Differentiable Marching Cubes (DiffMC) implementation utilizing CUDA, delivering smooth surfaces from density fields. It runs around $10\times$ faster than the previous prevalent mesh extraction algorithm (DMTet) at similar triangle counts.

- Furthermore, our mesh-based representation can be seamlessly integrated with GLSL shaders, enabling real-time rendering applications.

## 2 RELATED WORK

**Neural field representations.** Neural rendering methods have demonstrated photo-realistic scene reconstruction and rendering quality. In particular, NeRF (Mildenhall et al., 2021) introduced the neural radiance field representation and achieved remarkable visual quality with volume rendering techniques. Various neural field representations have been proposed for better efficiency and quality, including MipNeRF (Barron et al., 2021) and RefNeRF (Verbin et al., 2022) that are based on coordinate-based MLPs, TensoRF (Chen et al., 2022a) and DiF(Chen et al., 2023a) that leverage tensor factorization, iNGP (Müller et al., 2022) that introduces multi-scale hashing, Plenoxels(Fridovich-Keil et al., 2022) and DVGO(Sun et al., 2022) that are voxel-based, and Point-NeRF (Xu et al., 2022) that is based on neural point clouds.

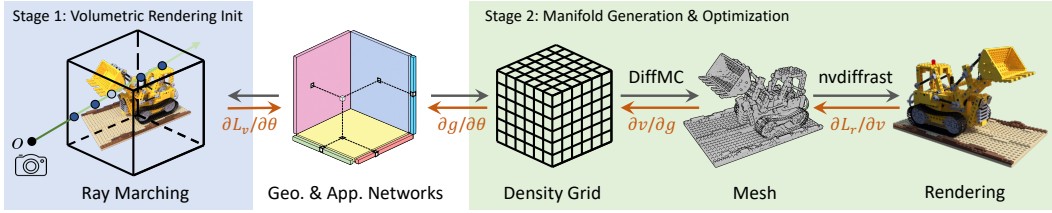

Fig. 2. Overall training pipeline for Stage 1 and 2 of NeuManifold. In Stage 1, volumetric rendering pipelines are used to initialize geometry and appearance networks. In Stage 2, the initialized geometry and appearance networks are further trained in differentiable rasterization with the help of DiffMC. The generated watertight manifold mesh and optimized appearance network are used in deployment.

However, most neural field representations represent 3D geometry as a volume density field, which is hard to edit for 3D applications other than rendering. While several methods have enabled appearance editing or shape deformation for such neural fields (Xiang et al., 2021; Zhang et al., 2022a; Yuan et al., 2022; Wu et al., 2022; Kuang et al., 2022; Chong Bao and Bangbang Yang et al., 2022), it is still highly challenging to apply them directly in modern 3D engines. Recent methods have proposed replacing density fields with volume SDFs to achieve better surface reconstruction with volume rendering (Wang et al., 2021; 2022; Yariv et al., 2021; Oechsle et al., 2021). However, neither density- or SDF-based models can be easily exported as meshes without significantly losing their *rendering* quality. Another recent work MobileNeRF (Chen et al., 2022b) converts the neural field into a triangle soup for real-time rendering. However, their mesh does not model accurate scene geometry and thus cannot be used for downstream applications. Our work offers a general solution to convert volumetric neural field representations to high-quality manifold meshes, enabling both high-quality rendering and broad additional 3D applications like physical simulation.

**Mesh reconstruction and rendering.** Polygonal meshes are a staple in modern 3D engines, widely employed for modeling, simulation, and rendering. Previous research has extensively explored mesh reconstruction from multi-view captured images through photogrammetry systems(Pollefeys & Gool, 2002; Snavely et al., 2006; Schönberger et al., 2016) like structure from motion(Schönberger & Frahm, 2016; Tang & Tan, 2019; Vijayanarasimhan et al., 2017), multi-view stereo(Furukawa & Ponce, 2010; Kutulakos & Seitz, 2000; Schönberger et al., 2016; Yao et al., 2018; Cheng et al., 2020), and surface extraction techniques(Lorensen & Cline, 1987; Kazhdan et al., 2006). However, achieving photorealistic rendering with classical photogrammetry pipelines remains a formidable challenge.

On the other hand, inverse rendering aims to fully disentangle intrinsic scene properties from captured images (Goldman et al., 2009; Hernandez et al., 2008; Zhang et al., 2021b; Bi et al., 2020b;c;a; Zhang et al., 2021a; Li et al., 2018; Zhang et al., 2022b;c). Recent methods, such as nvdiffrec (Munkberg et al., 2021), nerf2mesh(Tang et al., 2022), BakedSDF (Yariv et al., 2023), achieve high-quality reconstruction and fast rendering speed. Nevertheless, these methods often introduce self-intersections or an excessive number of triangles in the mesh reconstruction, which are undesired for simulation and geometry processing tasks.

Moreover, in recent years, several studies (Liao et al., 2018; Remelli et al., 2020; Shen et al., 2021; Mehta et al., 2022; Shen et al., 2023) have delved into differentiable mesh extraction algorithms due to their crucial role in mesh optimization workflows, connecting implicit field and explicit mesh representations. While they still have limitations, such as surface artifacts (Shen et al., 2021) and a lack of manifold guarantees (Liao et al., 2018; Shen et al., 2023). Our DiffMC generates significantly smoother surfaces on density fields and maintains watertight manifold properties. In summary, our approach leverages neural field reconstruction to provide a high-quality initialization for differentiable rendering, and we integrate it with differentiable marching cubes to ensure that our final output is manifold and watertight. This enables our model to be directly applied to a wide range of 3D applications.

## 3 METHOD

We present a 3D reconstruction pipeline that reconstructs scene geometry and appearance from captured multi-view images. Our method consists of two main stages: initialization with differentiable

volume rendering and manifold generation with differentiable rasterization, as illustrated in Fig. 2, plus an optional fine-tuning stage. In particular, we leverage neural field representations with volume rendering-based reconstruction to offer the initialization for the subsequent mesh optimization, where we further optimize the *topology*, *geometry* and *appearance* with differentiable marching cubes and rasterization. Optionally, when the manifold property is not required, we fine-tune the *geometry* and *appearance* by directly moving mesh vertices. Finally, we deploy the pipeline with GLSL shaders for cross-platform real-time rendering and demonstrate the important role of anti-aliasing on visual quality.

### 3.1 NEURAL FIELD REPRESENTATION.

We represent a 3D scene with a geometry network $G$ and an appearance network $A$. In particular, given an arbitrary 3D location $x$, the geometry network outputs its corresponding volume density $\sigma$, and the appearance network regresses a view-dependent color $c$ at the location. This can be expressed by:

$$\sigma_x, c_x = G(x), A(x, d) \tag{1}$$

where $d$ is the viewing direction. Our approach supports any common neural field representations for the geometry and appearance networks. In this work, we choose the state-of-the-art neural field representation TensoRF (Chen et al., 2022a) as the network architecture.

In order to balance rendering quality and inference speed, we propose two kinds of appearance networks. For the *high-quality* version, we adopt Vector-Matrix (VM) decomposition plus an MLP. For the *fast* version, we utilize VM decomposition plus Spherical Harmonics (SH). This can greatly accelerate inference speed on deployment; we discuss this in detail in Sec. 3.5.

### 3.2 INITIALIZATION BY VOLUME RENDERING (STAGE 1)

In the first stage, we train the networks through differentiable volume rendering to establish a strong initialization for the subsequent differentiable rasterization-based optimization phase. As in NeRF, we render pixel colors $C$ using the volume density and view-dependent colors from our geometry and appearance models as:

$$C = \sum_{i=1}^{N} T_i(1 - \exp(-\sigma_i \delta_i)) c_i, \quad T_i = \exp(-\sum_{j=1}^{i-1} \sigma_j \delta_j). \tag{2}$$

where $T$ is the volume transmittance and $\delta$ is the ray marching step size. This differentiable rendering process allows us to optimize our networks with a rendering loss.

### 3.3 MANIFOLD GENERATION & OPTIMIZATION (STAGE 2)

In the second stage of our process, we leverage a similar pipeline as nvdiffrec (Munkberg et al., 2021) to optimize the object topology, geometry and appearance simultaneously. Unlike nvdiffrec that directly optimizes the SDF function from scratch, we utilize the pre-trained TensoRF models from the previous stage as initialization. Additionally, we replace the marching algorithm from Differentiable Marching Tetrahedra (DMTet) (Shen et al., 2021) with our Differentiable Marching Cubes (DiffMC), which seamlessly integrates pre-trained density networks into the differentiable rasterization pipeline and significantly reduces artifacts on mesh surfaces.

Different from nvdiffrec, which optimizes SDF values stored on the grid, our methods need to convert the output of the density network to these values, since SDF-based methods tend to exhibit lower visual fidelity and can lose high-frequency details in the geometry. With the

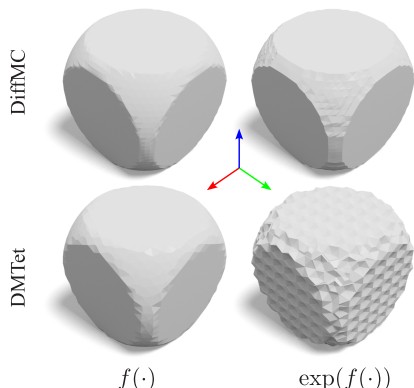

Fig. 3. DiffMC vs DMTet on non-linear fields. $f(\cdot)$ represents the SDF function and $\exp()$ is a non-linear transformation.

pre-trained TensoRF density network, we convert their density into opacity as: $\alpha = 1 - \exp(-\sigma \cdot \delta)$, where $\sigma$ denotes density, $\alpha$ denotes opacity, $\delta$ is the ray step size used in volume rendering. We

consider a threshold $t$ that controls the position of the surface with respect to opacity and send the value $\alpha - t$ to DiffMC to obtain our manifold mesh.

During the conversion of the density field into a mesh, we encountered notable artifacts with DMTet. These issues arose primarily from the non-linear nature of the density field. As illustrated in Fig. 3, when we apply a non-linear transformation like $\exp()$ to a standard SDF field, the mesh extracted by DMTet develops deformations with peaks and valleys. This happens because the conventional linear interpolation method is no longer adequate for dealing with the non-linear field, and the way it divides space into tetrahedra results in artifacts on surfaces that don't align well with these tetrahedral divisions. Given that most real-world objects tend to be axis-aligned, adopting an axis-aligned space division can significantly reduce these artifacts. More explanations are in Appendix Sec. B.

Therefore, we introduce Differentiable Marching Cubes (DiffMC), which operates on an axis-aligned grid. DiffMC not only extracts the mesh using the conventional marching cubes algorithm (Lorensen & Cline, 1998) but also provides vertex gradients with respect to the grid, denoted as $\frac{\partial v}{\partial g}$. This enables the mesh extraction process to be seamlessly combined with the mesh optimization pipeline using the chain rule: $\frac{\partial L}{\partial \theta} = \sum_{v \in V} \frac{\partial L}{\partial v} \frac{\partial v}{\partial g} \frac{\partial g}{\partial \theta}$, where $L$ is the rendering loss, $\theta$ is the parameters in the density network, $V$ is the set of mesh vertices and $g$ is the grid. In a manner akin to the approach outlined in Shen et al. (2021), we incorporate deformable vectors into the grid that can be optimized. This allows the extracted mesh to adjust more effectively to the desired shape by making subtle adjustments within half of the cube. As shown in Fig. 3, DiffMC is less influenced by the non-linearity and is capable of producing significantly smoother surfaces, even on geometries that are not aligned with the axis. To our best knowledge, we are the first to implement the complete differentiable marching cubes, achieving exceptionally fast speeds that are $10\times$ faster than DMTet.

We put the resulting mesh into nvdiffrast (Laine et al., 2020) to render 2D images and use the rendering loss to update the geometry and appearance networks. Precisely, the points on the mesh surface are passed through the appearance network to generate the output color for each pixel.

With a strong initialization from networks pre-trained in volume rendering and the marching cubes algorithm, we are able to get watertight manifold meshes that are more accurate than both volumetric rendering and mesh rendering alone, with better visual quality.

### 3.4 GEOMETRY AND APPEARANCE FINETUNE (STAGE 3)

The mesh generated in the previous stage is guaranteed to be a watertight manifold, which satisfies the rigorous requirements of common geometry processing algorithms. However, maintaining manifoldness may come at the cost of rendering quality, particularly for areas with intricate structures where preserving both structural details and optimal triangular connections can be challenging.

We address this issue with an optional fine-tuning stage to further enhance rendering quality for applications where manifold properties are not necessary. Here, we solely fine-tune the mesh vertex positions and appearance network to improve the rendering loss. While this operation may introduce self-intersections, it preserves the original good edge connections, thus retaining watertightness.

### 3.5 DEPLOYMENT

**GLSL shaders.** Our pipeline produces a triangle mesh with an appearance network consisting of TensoRF and MLPs. This can be directly mapped to a traditional rasterization pipeline as GLSL shaders. We upload TensoRF weights as three 3D textures and three 1D textures with linear filtering, and MLP weights as 2D textures. After rasterizing triangles in the vertex shader, we evaluate TensoRF and MLPs in the fragment shader with model-space coordinates and viewing directions.

We further accelerate the deployed rendering pipeline using different MLP size models as well as a spherical harmonics version. We summarize these quality/speed trade-offs in Table 5.

**Anti-Aliasing.** Aliasing is a common issue in rasterization pipelines due to the undersampling of high-frequency features, such as mesh edges and detailed textures. In contrast to volumetric rendering, where semi-transparent volumes can mitigate aliasing, mesh-based rendering pipelines are significantly affected by this problem.

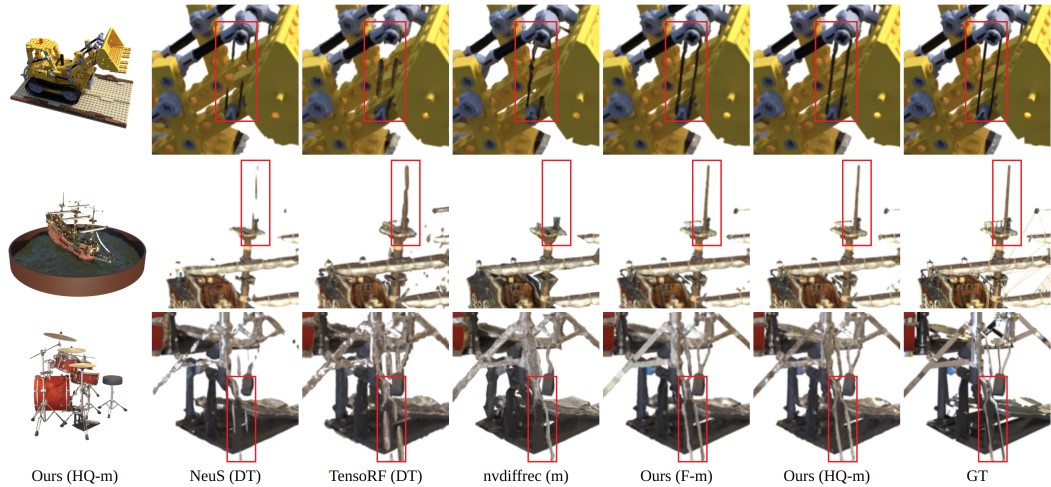

| Ours (HQ-m) | NeuS (DT) | TensoRF (DT) | nvdiffrec (m) | Ours (F-m) | Ours (HQ-m) | GT |

Fig. 4. Rendering quality comparison between our and existing mesh rendering methods. Our methods are able to well preserve thin structures as well as achieve high rendering quality. (DT: direct transfer; F: fast; HQ: high-quality; m: manifold).

Supersample anti-aliasing (SSAA) is the most straightforward method to mitigate aliasing; it renders high-resolution images and down-samples them to the target resolution. While SSAA provides the best visual quality, it is computationally expensive due to the increased resolution. An alternative approach is multisample anti-aliasing (MSAA), which is enabled by modern GPU hardware. MSAA reduces the cost of anti-aliasing by increased shader evaluation only on pixels covered by multiple triangles, and it shades each triangle once. It improves visual fidelity at a relatively small performance hit, as shown in Appendix Fig. 16.

## 4 EXPERIMENTS

### 4.1 IMPLEMENTATION DETAILS

For the first stage, we directly build on off-the-shelf volume rendering models. Specifically, for TensoRF, we use the official implementation. We compare two of our models for our main results: a high-quality one, labeled with Ours (HQ), which uses the TensoRF (VM) with 48-dim input features and 12-dim output features, plus a three-layer MLP decoder; a fast one, labeled with Ours (F) that uses the TensoRF (VM) with 48-dim input features and output 27-dim SH coefficients. More details about the network architecture are in supplementary.

We train all the stage 2 and 3 models with batch size of 2 for 10k iterations. We use DiffMC with a grid resolution of 256 for all results. Except when comparing with nvdiffrec, we use the default resolution of 128 as nvdiffrec's performance drops on higher resolutions, possibly due to the decreased batch size and harder optimization.

### 4.2 COMPARISON ON NOVEL-VIEW SYNTHESIS

In Table 1, we show a quantitative comparison on novel view synthesis between our method and other neural rendering and differentiable rasterization methods. We perform the experiments on the widely-used NeRF-Synthetic dataset (Mildenhall et al., 2021). We observe that even though NeuS and TensoRF have very high quality using their original volume rendering, when directly transferred to mesh rendering without any fine-tuning—shown as NeuS (DT) and TensoRF (DT) in the table— they have sharp performance drops. Specifically, we essentially extract meshes from their density or SDF fields and then use them for surface rendering. This involves fetching color information from their appearance networks using surface points. Nvdiffrec can generate watertight and manifold meshes but its rendering quality has a large gap with other neural rendering methods. In contrast, our models (both high-quality and fast) can achieve high quality on both mesh reconstruction and rendering. It is worth noting that our method attains the highest rendering quality compared to all

| Method | Geometry | Mesh | Watertight | Manifold | PSNR↑ | SSIM↑ | LPIPS↓ |
|--------|----------|------|-----------|----------|-------|-------|--------|
| NeRF | Volume | ✗ | - | - | 31.00 | 0.947 | 0.081 |
| TensoRF | Volume | ✗ | - | - | 33.20 | 0.963 | 0.050 |
| NeuS* | Volume | ✗ | - | - | 30.74 | 0.951 | 0.064 |
| MobileNeRF | Mesh | ✓ | ✗ | ✗ | 30.90 | 0.947 | 0.062 |
| nvdiffrec | Mesh | ✓ | ✓ | ✗ | 28.90 | 0.938 | 0.073 |
| nerf2mesh | Mesh | ✓ | ✓ | ✗ | 29.76 | 0.940 | 0.072 |
| TensoRF (DT) | Mesh | ✓ | ✓ | ✓ | 25.28 | 0.886 | 0.115 |
| NeuS* (DT) | Mesh | ✓ | ✓ | ✓ | 27.85 | 0.935 | 0.074 |
| nvdiffrec (m) | Mesh | ✓ | ✓ | ✓ | 27.65 | 0.933 | 0.084 |
| Ours (F) | Mesh | ✓ | ✓ | ✗ | 30.94 | 0.952 | 0.061 |
| Ours (HQ) | Mesh | ✓ | ✓ | ✗ | **31.65** | **0.956** | **0.056** |
| Ours (F-m) | Mesh | ✓ | ✓ | ✓ | 30.47 | 0.949 | 0.065 |
| Ours (HQ-m) | Mesh | ✓ | ✓ | ✓ | 31.19 | 0.954 | 0.059 |

Table 1. Average results on NeRF-Synthetic dataset. The results of NeRF, MobileNeRF and nerf2mesh are taken from their papers, and the other results for mesh rendering are tested on our machine using Pytorch implementation. (DT: direct transfer) * instant-nsr-pl Guo (2022) implementation. The **geometry property** is grouped by color.

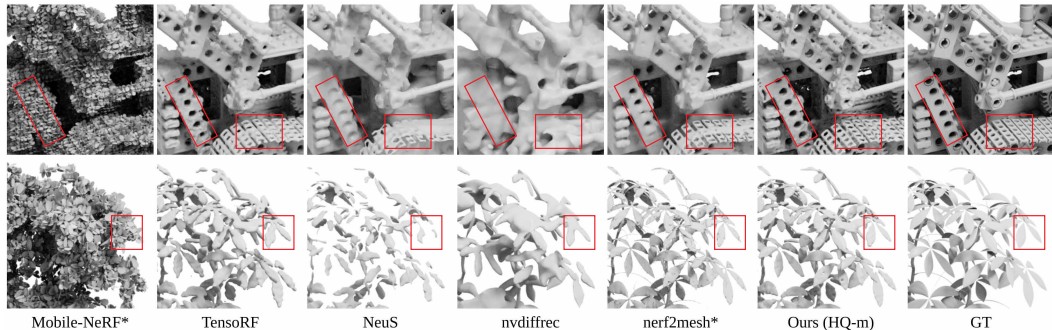

Mobile-NeRF*  TensoRF  NeuS  nvdiffrec  nerf2mesh*  Ours (HQ-m)  GT

Fig. 5. Visual comparison of mesh quality of different methods. MobileNeRF generates a triangle soup that can only preserve the rough shape. Nvdiffrec gives coarse mesh and fails on some regions. TensoRF is over-detailed and NeuS is over-smoothed. Our method combines the merits of these methods, which is comparable and even better than non-manifold meshes of nerf2mesh. (Non-manifold methods are denoted by *)

other surface rendering techniques, surpassing even those that generate non-manifold meshes. In addition, when the manifold property is not required, our visual fidelity can be further boosted with the third stage fine-tuning, leading to an average PSNR 0.65 dB higher than vanilla NeRF.

We also show visual comparisons on mesh-based rendering in Fig. 4. We can clearly see that the mesh rendering with meshes directly extracted from NeuS and TensoRF fails to recover the high-frequency details and thin structures in the scene. Moreover, since they apply volume rendering and integrate the colors of multiple points along the ray to match the training images during the optimization, simply extracting the color of a single point at the isosurface cannot faithfully recover the appearance of the scene. Nvdiffrec directly applies mesh rendering during the training, but the recovered

| Method | Geo. | Outdoor | Indoor |
|--------|------|---------|--------|
| NeRF | Vol | 21.46 | 26.84 |
| NeRF++ | Vol | 22.76 | 28.05 |
| mip-NeRF | Vol | 24.47 | 31.72 |
| Mobile-NeRF | Mesh | 21.95 | - |
| BakedSDF | Mesh | **22.47** | 27.06 |
| Ours (HQ-m) | Mesh | 21.07 | 25.80 |
| Ours (HQ) | Mesh | 22.05 | **27.63** |

Table 2. PSNR of Unbounded scenes. More metrics are in the Appendix.

meshes can miss complex structures, thus resulting in a degradation in visual quality. In contrast, our method benefits from the initialization from the neural volume rendering and can better recover the fine-grained details of the scene.

We also verify the effectiveness of our method on two real datasets, MipNeRF-360 dataset (Barron et al., 2022) and LLFF dataset (Mildenhall et al., 2019). The quantitative results on MipNeRF-360 is shown in Table 2, where our method significantly outperforms other mesh-based methods on indoor scenes. More results are in the Appendix.

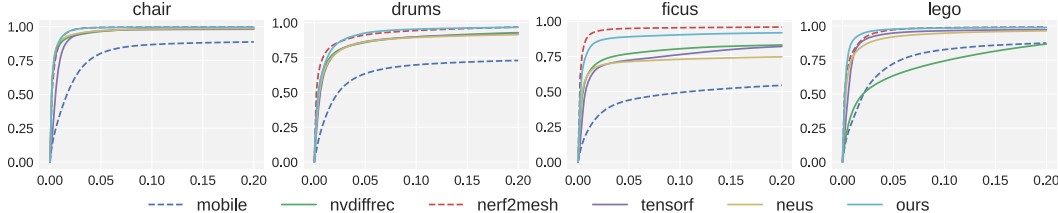

Fig. 6. Average VSA-tolerance plot for test views from four NeRF-Synthetic scenes. Depth map of the mesh produced by Ours (HQ-m) achieves high matching scores consistently for different tolerance values. (Non-manifold methods are denoted by dotted line)

| G. Init | A. Init | PSNR↑ | SSIM↑ | LPIPS↓ |
|---|---|---|---|---|
| ✗ | ✗ | 20.56 | 0.826 | 0.204 |
| ✗ | ✓ | 24.43 | 0.882 | 0.149 |
| ✓ | ✗ | 29.74 | 0.945 | 0.067 |
| ✓ | ✓ | **31.19** | **0.954** | **0.059** |

Table 3. Ablation study for Stage 1. Using initializations from volume rendering enables more accurate mesh reconstruction and rendering, leading to more accurate novel view synthesis.

| Geo. + App. | PSNR↑ | SSIM↑ | LPIPS↓ |
|---|---|---|---|
| GT + TF | 31.78 | 0.958 | 0.053 |
| TFmesh + MLP | 26.28 | 0.915 | 0.203 |
| TFmesh + Hash | 26.62 | 0.921 | 0.090 |
| TFmesh + SH | 26.48 | 0.909 | 0.103 |
| TFmesh + TF | 27.00 | 0.929 | 0.081 |
| TFmesh (opt) + TF | **29.74** | **0.945** | **0.067** |

Table 4. Ablation study for Stage 2. Our full method (last row) jointly optimizes geometry and appearance and achieves the best performance. (TF: TensoRF)

### 4.3 COMPARISON ON MESH RECONSTRUCTION

We observe that traditional mesh-distance metrics such as Chamfer distance are not suitable for mesh quality comparison, as they are often dominated by the performance of regions unseen during training. To this end, we propose to use the visible surface agreement (VSA) metric, modified from the visible surface discrepancy proposed by Hodaň et al. (2020):

$$e_{\text{VSA}} = \underset{p \in V \cup \hat{V}}{\text{avg}} \begin{cases} 1 & \text{if } p \in V \cap \hat{V} \wedge |D(p) - \hat{D}(p)| < \tau \\ 0 & \text{otherwise} \end{cases}$$

where given a view, $D$ and $\hat{D}$ denote the depth map of the ground-truth and reconstructed meshes, $V$ and $\hat{V}$ denote the pixel visibility masks, and $\tau$ is the misalignment tolerance. Higher VSA indicates a better match between depth maps.

We compare the average VSA metric over 200 testing views of the NeRF-Synthetic dataset with different misalignment tolerances in Fig. 6 (others in Appendix Fig. 18). We additionally provide a visual comparison of the reconstructed meshes in Fig. 5. From the comparisons, we can clearly see that our method achieves consistently better VSA performance than the manifold mesh baseline methods. Our generated meshes better capture the detailed structures of the scene such as the lego wheels, even better than nerf2mesh (Tang et al., 2022) that generates non-manifold ones.

### 4.4 ABLATION FOR STAGE 1 & 2

To show the importance and effectiveness of using initialization from volume rendering, we design an ablation study using different initialization methods for Stage 1. As we can see from Table 3, directly optimizing the mesh without initialization from volume rendering, leads to the worst novel view synthesis performance. Both geometry initialization and appearance initialization can boost the accuracy, with geometry initialization playing a more critical role in the performance improvement. Additional visual results in the Appendix Fig. 17 highlight that when high-resolution grids are employed without appropriate geometry initialization, the mesh optimization process can easily become trapped in a local minimum.

We validate the necessity of optimizing the meshes in Table 4. To achieve this, we compare against baselines that keep the meshes from Stage 1 fixed and only optimize the appearance.We also provide the results using the GT mesh in combination with the TensorF appearance network as a reference, representing the upper limit of texture optimization methods. As we can see from the results, using the meshes without further optimization achieves much lower accuracy than our full method,

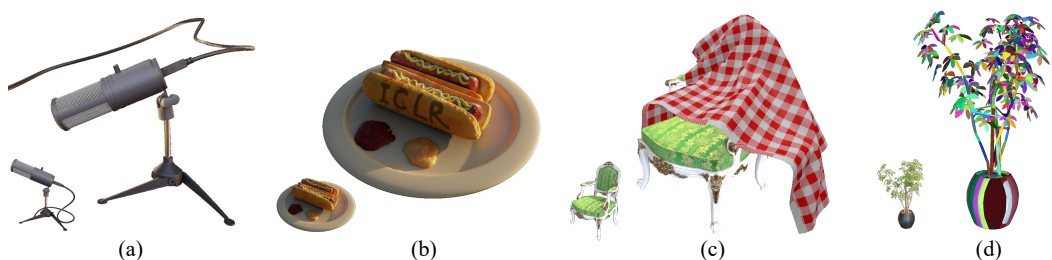

(a)       (b)       (c)       (d)

Fig. 7. Applications of NeuManifold. (a) Geometry editing with Laplacian surface editing. (b) Appearance editing with vertex painting. (c) Collision shape for cloth simulation. (d) Collision-aware convex decomposition.

which demonstrates the essential of jointly optimizing the geometry and appearance in Stage 2. All appearance networks were trained from scratch for fair comparison.

### 4.5 SPEED AND QUALITY TRADE-OFF

We show the model performance and speed after being deployed into GLSL in Table 5 and show the trade-off between model capacity and inference speed. FPS is computed with the average time to render the first frame in the test set of NeRF-Synthetic dataset on an NVIDIA RTX 4090.

| Params | AA | PSNR↑ | SSIM↑ | LPIPS↓ | FPS |
|---|---|---|---|---|---|
| #feat=48 | 8× MS | 30.34 | 0.949 | 0.062 | 93 |
| mlp=3×64 | 16× SS | 31.16 | 0.954 | 0.057 | 26 |
| #feat=48 | 8× MS | 29.73 | 0.942 | 0.071 | 322 |
| mlp=3×16 | 16× SS | 30.49 | 0.947 | 0.064 | 86 |
| #feat=12 | 8× MS | 30.11 | 0.946 | 0.066 | 98 |
| mlp=3×64 | 16× SS | 30.90 | 0.951 | 0.060 | 27 |
| #feat=12 | 8× MS | 29.55 | 0.941 | 0.073 | 585 |
| mlp=3×16 | 16× SS | 30.28 | 0.946 | 0.066 | 163 |
| #feat=48 | 8× MS | 29.73 | 0.943 | 0.068 | 312 |
| SH | 16× SS | 30.44 | 0.949 | 0.063 | 82 |

Table 5. Trade-off between rendering speed and quality with different appearance network capacity. 8× MS: 8× sample per-pixel MSAA, 16× SS: 16× sample per-pixel SSAA.

## 5 APPLICATIONS

With a manifold mesh-based geometry representation, NeuManifold can be easily plugged into a wide variety of 3D content creation tools. This is a significant advantage over previous neural reconstruction methods and we demonstrate three such applications below.

**Geometry editing.** Geometry editing algorithms often rely on good input mesh connectivity. In Fig. 7a, we demonstrate Laplacian surface editing Sorkine et al. (2004) for non-rigid deformation of the reconstructed microphone.

**Appearance editing.** Our meshes integrate directly into modeling software and can be edited by artists. In Fig. 7b, we load the generated mesh into Blender and paint its vertices. The painted color is multiplied with the original color in the GLSL shader.

**Physical Simulation.** Our reconstructed meshes can be used as static collision meshes for soft-body simulation (e.g., cloth simulation as shown in Fig. 7c) similar to previous works. Moreover, the watertight and manifold properties enable a wider range of applications. For example, they can be used as direct input to the collision-aware convex decomposition algorithm (Wei et al., 2022) for rigid-body collision shape generation (Fig. 7d). They can be directly converted to finite-element meshes by Delaunay tetrahedralizations (Hang, 2015) and used in a finite-element simulation with incremental potential contact (IPC) (Li et al., 2020) ( Fig. 1 and Appendix Fig. 12).

## 6 CONCLUSION

We have introduced a novel method for reconstructing high-quality, watertight manifold meshes with accurate rendering support from multi-view images. However, our method currently faces limitations when dealing with specular areas, like the "materials" in NeRF-Synthetic and the "room" in the LLFF dataset. In these cases, the reconstructed meshes may exhibit discontinuities to capture the effect of different colors for the same point seen from different views. We believe that addressing this issue will require the incorporation of inverse rendering techniques and the inclusion of additional priors to ensure a more accurate geometry.

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

## A PRELIMIARIES

### A.1 WATERTIGHT AND MANIFOLD MESHES

**Watertight.** If all edges are shared by exactly two faces, then the mesh is watertight.

**Manifold.** A manifold mesh must meet the following properties: (1) all edges must connect at most two faces; (2) each edge is incident to one or two faces and faces incident to a vertex must form a closed or open fan; (3) the faces must not self-intersect with each other.

### A.2 VOLUMETRIC NEURAL FIELDS.

Recent neural field representations utilize differentiable volume rendering for their reconstruction and leads to high visual quality. While our approach can generally support any neural field models, we apply two specific ones, TensoRF and NeuS, in our paper. We now briefly cover the preliminaries of these methods.

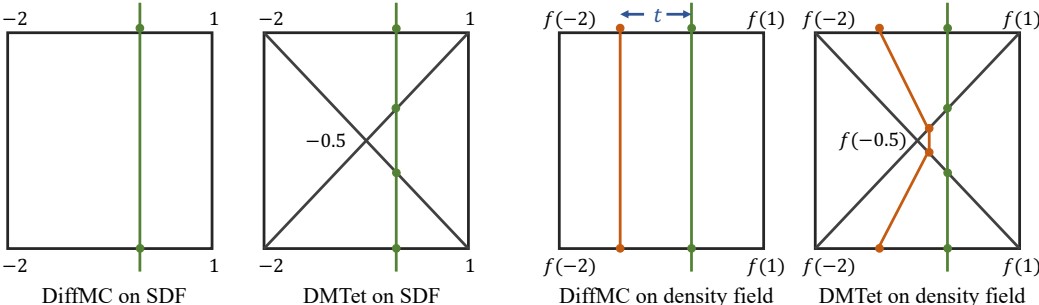

Fig. 8. 2D example illustrating why DMTet tends to introduce more artifacts when extracting meshes from density fields while DiffMC can generate much smoother surfaces.

**TensoRF.** The original NeRF uses pure MLPs, which make it slow to train and incapable of modeling details accurately. TensoRF (Chen et al., 2022a) decodes the radiance field from a volume of features, and this feature volume is further factorized into factors leveraging CANDECOMP/PARAFAC decomposition or vector-matrix decomposition. In this work, we are interested in the vector-matrix decomposition, which factorizes the 4D feature volume as the sum of three outer products between a matrix and a vector.

### A.3 DIFFERENTIABLE RASTERIZATION

Differentiable rasterization refers to methods that optimize inputs of rasterization-based rendering pipelines. In this work, we are interested in nvdiffrast (Laine et al., 2020), which consists of 4 stages, rasterization, interpolation, texture lookup, and anti-aliasing. We mainly use the rasterization stage, which maps triangles from 3D space onto pixel space, and the interpolation stage, which provides 3D coordinates of pixels to query the appearance network.

To ensure the mesh optimized by differentiable rasterization is a watertight manifold, we need to apply a meshing algorithm that generates such meshes. In this work we propose DiffMC, which divides the 3D space into a deformable grid and takes a scalar field (often SDF) defined on its vertices as input. The algorithm turns the scalar field into an explicit mesh by a differentiable marching cubes algorithm.

## B DIFFERENTIABLE MARCHING CUBES (DIFFMC)

In this section, we present additional results for DiffMC. These include a 2D example showing why DMTet tends to introduce more artifacts on density fields than DiffMC, an ablation study that demonstrates how grid resolution influences visual fidelity and a comparison highlighting the effectiveness of our method in mesh reconstruction when compared to DMTet (Shen et al., 2021).

First, we illustrate how DMTet and DiffMC generate surfaces with a 2D schematic diagram in Fig. 8. In 2D, Marching Cubes is analogous to "Marching Squares" and Marching Tetrahedra is analogous to "Marching Triangles". Given a surface (shown as a green vertical line) passing through the square/triangle grids (shown as black lines), suppose we have recorded the perfect signed distance function (SDF) values of the surface on the grid nodes, as shown in the two leftmost figures, regardless of how the algorithm divides the space, both methods exactly recover the ground truth surface through linear interpolation.

However, in practice, perfect SDF values are not easily obtainable, especially when the input comes from a volumetric density representation. Here, we simulate an imperfect SDF by applying a nonlinear transformation $f(s) = \exp(s) - 1 - t$ to the SDF values. Under this scenario, DiffMC can still generate a flat surface (red line in the second figure from right), albeit with a slight offset $t$ which can be rectified by introducing an adjustable threshold to the grid values. In contrast, DMTet produces zigzag lines (red line in the rightmost figure) due to varying space divisions and cannot be easily fixed.

| DiffMC reso | 32 | 64 | 100 | 128 | 200 | 256 | 300 | 384 | 400 |
|---|---|---|---|---|---|---|---|---|---|
| PSNR | 23.12 | 26.83 | 28.64 | 29.46 | 30.8 | 31.19 | 31.34 | 31.53 | 31.54 |
| SSIM | 0.894 | 0.925 | 0.94 | 0.946 | 0.952 | 0.954 | 0.955 | 0.956 | 0.956 |
| LPIPS | 0.121 | 0.089 | 0.075 | 0.069 | 0.061 | 0.059 | 0.057 | 0.056 | 0.056 |

Table 6. The influence of DiffMC resolution to rendering quality. The visual fidelity consistently improves as the resolution increases, eventually reaching a plateau when it reaches 400.

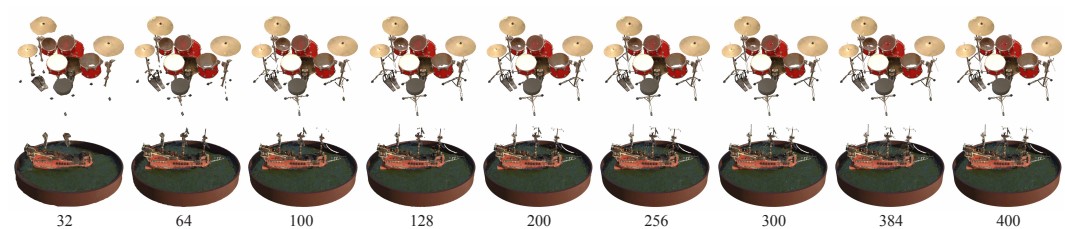

Fig. 9. The influence of DiffMC resolution to rendeirng quality. We have noticed that lower resolutions can capture most of the coarse structures but tend to lose finer details, such as the drum legs and the ropes on the ship. These finer details become more discernible as the resolution increases.

As we transition from lower to higher resolutions, we observe a consistent improvement in rendering quality, ultimately converging as the resolution reaches 400, as demonstrated in Table 6. Moreover, as depicted in Fig. 9, a higher-resolution DiffMC is notably more adept at recovering intricate structures, such as the ropes on the ship.

Next, we highlight the advantages of our method in extracting meshes from density fields by applying both our approach and DMTet (Shen et al., 2021) to a set of pre-trained density networks, including TensoRF (Chen et al., 2022a), instant-NGP (Müller et al., 2022) and vanilla NeRF (Mildenhall et al., 2021). By comparing the visible surface agreemen (VSA) of the reconstructed meshes, as illustrated in Fig. 10, we observe a consistent enhancement brought about by DiffMC across all methods. We also conduct a comparison between our DiffMC and DMTet in our pipeline, noting a significant improvement in surface smoothness with our method, which effectively mitigates most of the artifacts resulting from the non-linearity of the density field.

We compare the speed of DiffMC and DMTet by running both forward and backward processes for $1000\times$ and then compute the average speed.

## C  MIP-NERF 360 DATASET

We evaluate our method on unbounded real scenes in the Mip-NeRF 360 dataset (Barron et al., 2022). To deal with the unbounded background, we follow the contraction function proposed in Barron et al. (2022) to warp the far objects from their original space, $t$-space, into the contracted space $s$-space (a sphere with a radius of 1.2 in our setup). When generating the mesh, we apply DiffMC on the geometry network within $t$-space so that the mesh can be watertight manifold, otherwise the contraction may break the property. After getting the points on the mesh surface, we contract the points back to $s$-space to compute the color. Within the $t$-space, we utilize multiple resolutions for the entire scene, with a higher resolution (340) for the foreground and a lower resolution (56) for the background. To represent the distant background that falls outside the [-4, 4] box range, we employ a skybox. We use the anti-aliasing of nvdiffrast (Laine et al., 2020) for this dataset.

Our method generates watertight manifold foreground meshes. Therefore, we can apply simulation algorithms on the foreground objects, as shown in Fig. 12, where we apply soft-body simulation on the flower and use a solid ball to hit it.

In Table 7, we compared our method with others. Some mesh rendering methods, such as MobileNeRF (Chen et al., 2022b) and nerf2mesh (Tang et al., 2022), provided results for selected scenes, while our method worked effectively on all unbounded scenes, particularly excelling in indoor scenes. Visual results of our method are depicted in Fig. 13.

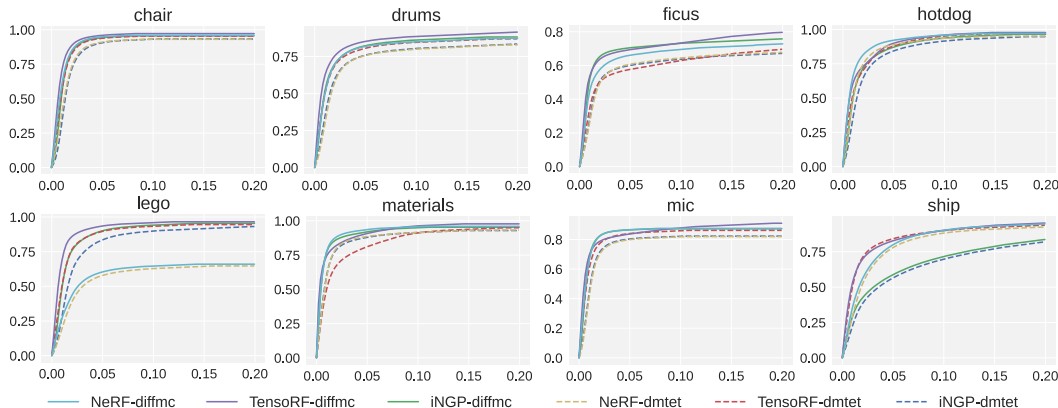

Fig. 10. DMTet vs DiffMC on extracting meshes from pre-trained density fields. Across all three methods, DiffMC consistently outperforms DMTet in terms of mesh quality.

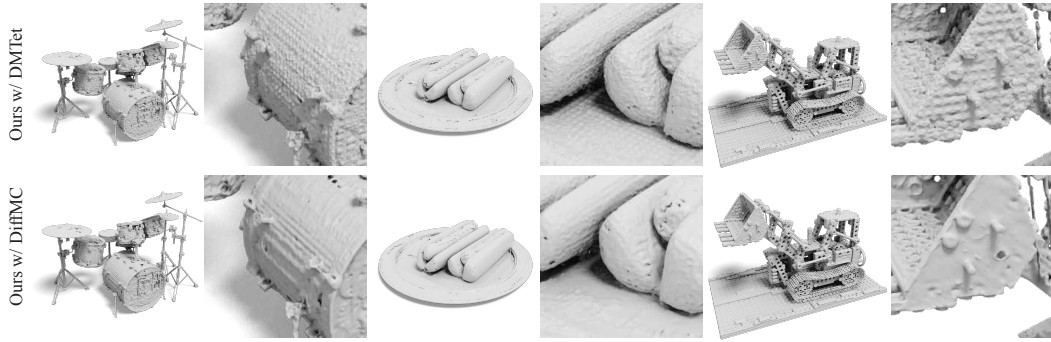

Fig. 11. A mesh surface comparison of Ours (HQ-m) between using DiffMC and DMTet reveals that DiffMC can create significantly smoother surfaces. This improvement is not limited to axis-aligned surfaces; it consistently outperforms DMTet on various rounded surfaces as well.

| PSNR | Bicycle | Garden | Stump | Flowers | Treehill | Bonsai | Counter | Kitchen | Room | Mean |
|---|---|---|---|---|---|---|---|---|---|---|
| MobileNeRF | **21.70** | 23.54 | **23.95** | **18.86** | **21.72** | - | - | - | - | - |
| nerf2mesh | 22.16 | 22.39 | 22.53 | - | - | - | - | - | - | - |
| BakedSDF | - | - | - | - | - | - | - | - | - | 24.51 |
| Ours (HQ-m) | 20.16 | 23.36 | 22.27 | 18.49 | 21.07 | 26.64 | 24.83 | 24.97 | 26.75 | 23.17 |
| Ours (HQ) | 21.38 | **24.90** | 23.51 | 18.82 | 21.64 | **28.61** | **26.31** | **26.63** | **28.95** | **24.53** |
| SSIM | | | | | | | | | | |
| MobileNeRF | 0.426 | 0.599 | 0.556 | 0.321 | 0.450 | - | - | - | - | - |
| nerf2mesh | **0.470** | 0.500 | 0.508 | - | - | - | - | - | - | - |
| BakedSDF | - | - | - | - | - | - | - | - | - | **0.697** |
| Ours (HQ-m) | 0.382 | 0.616 | 0.492 | 0.334 | 0.447 | 0.835 | 0.746 | 0.644 | 0.815 | 0.590 |
| Ours (HQ) | 0.469 | **0.746** | **0.589** | **0.366** | **0.494** | **0.888** | **0.808** | **0.764** | **0.872** | 0.666 |
| LPIPS | | | | | | | | | | |
| MobileNeRF | 0.513 | 0.358 | 0.430 | 0.526 | 0.522 | - | - | - | - | - |
| nerf2mesh | 0.510 | 0.434 | 0.490 | - | - | - | - | - | - | - |
| BakedSDF | - | - | - | - | - | - | - | - | - | **0.309** |
| Ours (HQ-m) | 0.561 | 0.372 | 0.475 | 0.553 | 0.560 | 0.268 | 0.346 | 0.380 | 0.348 | 0.429 |
| Ours (HQ) | **0.488** | **0.252** | **0.413** | **0.520** | **0.506** | **0.201** | **0.270** | **0.275** | **0.274** | 0.355 |

Table 7. Quantitative results on each scene in the Mip-NeRF 360 dataset.

# D    LLFF DATASET

We evaluate our method on forward-facing scenes on LLFF dataset (Mildenhall et al., 2019). Following Chen et al. (2022a), we contract the whole scene into NDC space to do the reconstruction

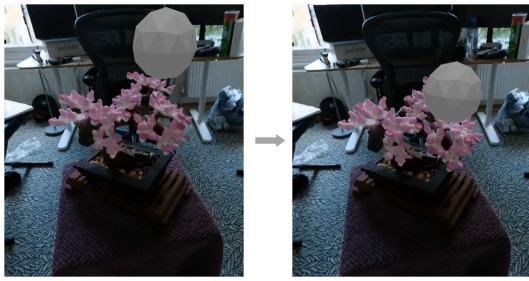

Fig. 12. Soft-body simulation on the foreground watertight manifold mesh. The solid ball hits the flower and makes it deform. See the project page for the full animation.

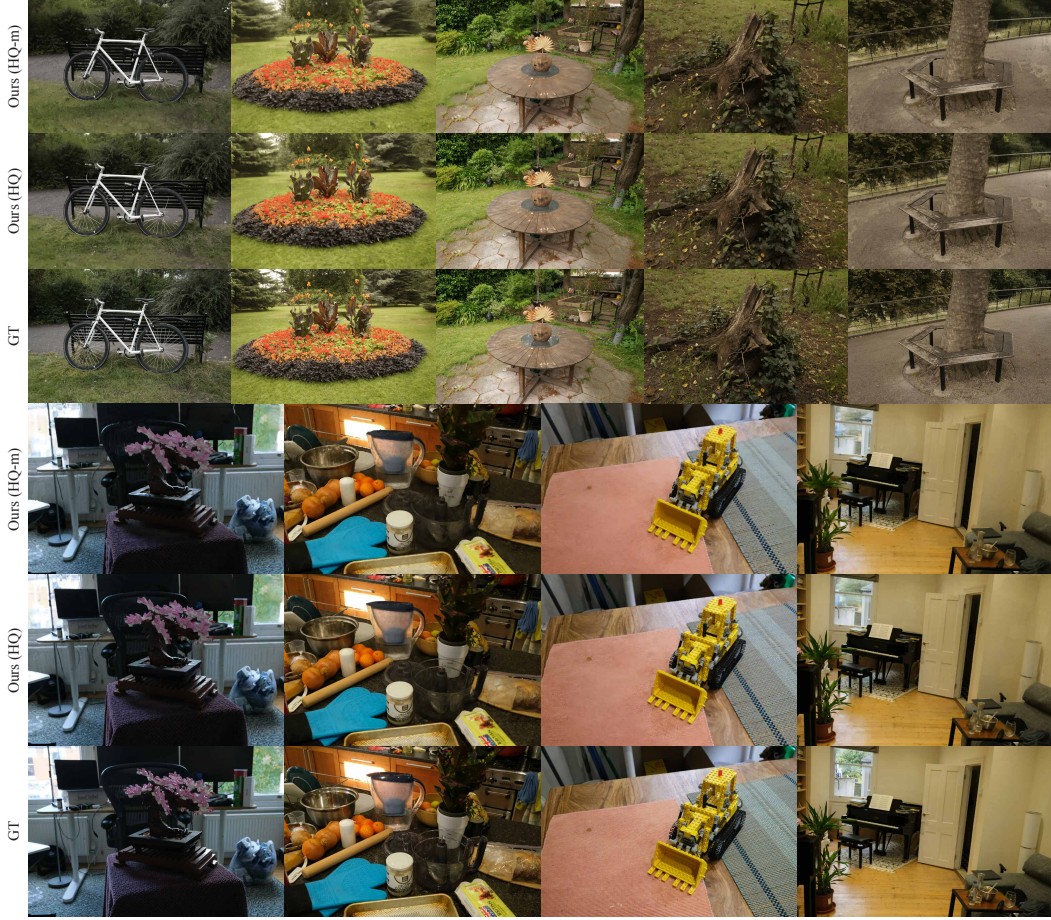

Fig. 13. Mip-NeRF 360 renderings.

and mesh extraction. On this dataset, we use DiffMC with resolution of 375. We use 9× sample per-pixel SSAA for this dataset. Table 8 and Fig. 14 shows the quantitative and qualitative results. Fig. 15 shows the reconstructed mesh of the scenes.

We put our method to the test with forward-facing scenes from the LLFF dataset (Mildenhall et al., 2019). In line with Chen et al. (2022a), we condensed the entire scene into NDC space for reconstruction and mesh extraction. For this dataset, we employed DiffMC with a resolution of 375. You can find both the quantitative results in Table 8 and the qualitative results in Fig. 14. Additionally, Fig. 15 showcases the reconstructed mesh for these scenes.

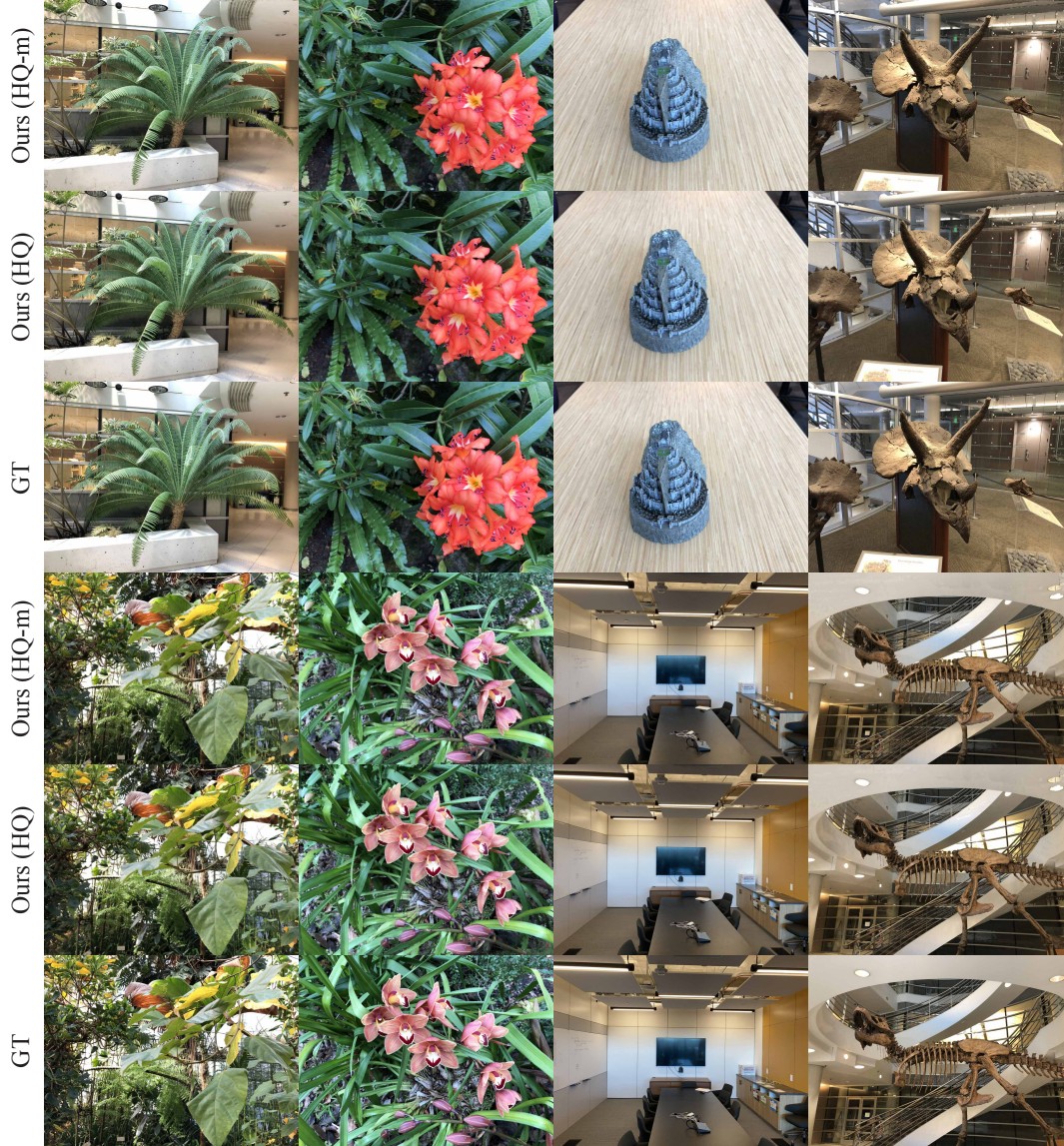

Fig. 14. LLFF renderings.

# E  NeRF-Synthetic Dataset

We show the complete quantitative comparison between our method and the previous works on the NeRF-Synthetic dataset in Table 9 and the complete visual comparison in Fig. 19.

# F  Mesh Quality

We show the mesh quality comparison in Fig.20, where except for Mobile-NeRF (Chen et al., 2022b) and nerf2mesh (Tang et al., 2022), all the meshes are watertight manifold. And we show the VSA-tolerance curves for the rest scenes in NeRF-Synthetic in Fig. 18.

# G  Network Architecture

In this section, we describe the network architecture used in the experiments. Our proposed method has two versions, a high-quality one and a fast one, and they share the same geometry network

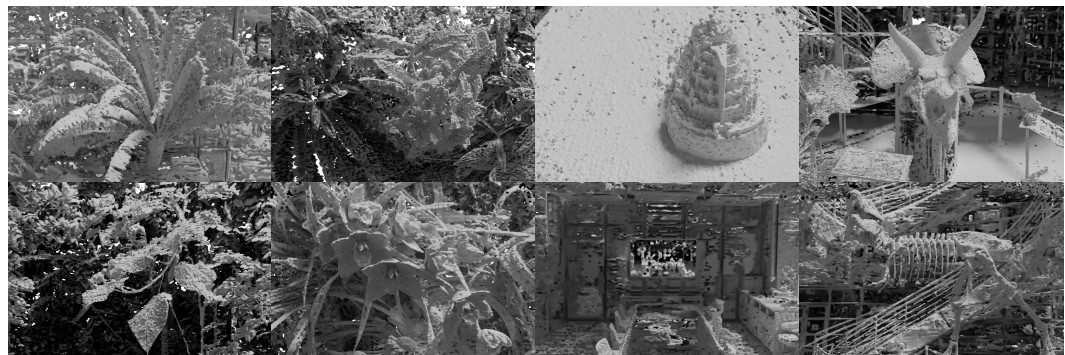

Fig. 15. LLFF mesh.

| PSNR | Fern | Flower | Fortress | Horns | Leaves | Orchids | Room | Trex | Mean |
|---|---|---|---|---|---|---|---|---|---|
| MobileNeRF | **24.59** | 27.05 | 30.82 | 27.09 | 20.54 | 19.66 | **31.28** | 26.26 | 25.91 |
| nerf2mesh | 23.94 | 26.48 | 28.02 | 26.25 | 19.22 | 19.08 | 29.24 | 25.80 | 24.75 |
| Ours (F-m) | 23.72 | 27.05 | 30.88 | 27.01 | 19.68 | 18.43 | 30.33 | 25.03 | 25.27 |
| Ours (F) | 24.05 | **27.22** | 30.98 | 27.09 | 19.92 | 18.91 | 30.63 | 25.58 | 25.55 |
| Ours (HQ-m) | 24.19 | 26.99 | 31.18 | 27.35 | 20.49 | 19.68 | 30.79 | 26.61 | 25.91 |
| Ours (HQ) | 24.54 | 27.08 | **31.32** | **27.49** | **20.59** | **19.73** | 31.11 | **27.16** | **26.13** |
| SSIM | | | | | | | | | |
| MobileNeRF | **0.808** | 0.839 | 0.891 | 0.864 | 0.711 | 0.647 | **0.943** | 0.900 | 0.825 |
| nerf2mesh | 0.751 | **0.879** | 0.765 | 0.819 | 0.644 | 0.602 | 0.914 | 0.868 | 0.780 |
| Ours (F-m) | 0.757 | 0.842 | 0.895 | 0.864 | 0.681 | 0.601 | 0.923 | 0.865 | 0.803 |
| Ours (F) | 0.772 | 0.848 | 0.898 | 0.866 | 0.693 | 0.622 | 0.926 | 0.872 | 0.812 |
| Ours (HQ-m) | 0.789 | 0.852 | **0.902** | 0.877 | 0.739 | 0.677 | 0.930 | 0.896 | 0.833 |
| Ours (HQ) | 0.801 | 0.856 | **0.902** | **0.881** | **0.745** | **0.681** | 0.933 | **0.904** | **0.838** |
| LPIPS | | | | | | | | | |
| MobileNeRF | **0.202** | 0.163 | **0.115** | 0.169 | 0.245 | 0.277 | **0.143** | **0.147** | **0.183** |
| nerf2mesh | 0.303 | 0.204 | 0.270 | 0.260 | 0.321 | 0.314 | 0.246 | 0.215 | 0.267 |
| Ours (F-m) | 0.274 | 0.181 | 0.158 | 0.196 | 0.254 | 0.278 | 0.208 | 0.256 | 0.226 |
| Ours (F) | 0.258 | 0.175 | 0.152 | 0.191 | 0.244 | 0.260 | 0.203 | 0.247 | 0.216 |
| Ours (HQ-m) | 0.245 | 0.164 | 0.137 | 0.171 | 0.202 | 0.234 | 0.188 | 0.216 | 0.195 |
| Ours (HQ) | 0.228 | **0.160** | 0.136 | **0.165** | **0.198** | **0.226** | 0.181 | 0.205 | 0.187 |

Table 8. Quantitative results on each scene in the LLFF dataset.

architecture but with different appearance networks. The geometry network is the same as TensoRF (Chen et al., 2022a) VM-192 in its paper. The appearance network is from TensoRF and we show the two versions below respectively.

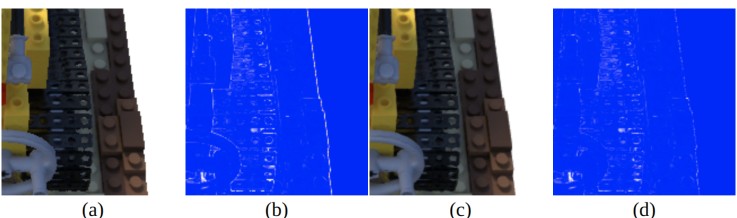

(a)  (b)  (c)  (d)

Fig. 16. Comparison between 8x MSAA and no AA. (a) Our deployed high-quality model without AA (FPS: 146, PSNR: 31.26). (c) the same model with $8\times$ MSAA (FPS: 93, PSNR: 33.01). (b) and (d) show the error maps of (a) and (c) respectively. The visual quality at edges is significantly improved by MSAA with a relatively small performance hit.

| PSNR | Chair | Drums | Ficus | Hotdog | Lego | Materials | Mic | Ship | Mean |
|---|---|---|---|---|---|---|---|---|---|
| MobileNeRF | 34.09 | 25.02 | 30.20 | 35.46 | 34.18 | 26.72 | 32.48 | 29.06 | 30.90 |
| nvdiffrec | 31.00 | 24.39 | 29.86 | 33.27 | 29.61 | 26.64 | 30.37 | 26.05 | 28.90 |
| TensoRF (DT) | 27.72 | 22.20 | 25.66 | 28.85 | 25.86 | 22.12 | 26.13 | 23.67 | 25.28 |
| NeuS (DT) | 31.80 | 22.52 | 23.44 | 33.86 | 28.07 | 26.68 | 31.42 | 25.02 | 27.85 |
| nerf2mesh | 31.93 | 24.80 | 29.81 | 34.11 | 32.07 | 25.45 | 31.25 | 28.69 | 29.76 |
| nvdiffrec (m) | 31.24 | 23.17 | 25.11 | 32.67 | 28.44 | 26.33 | 29.39 | 24.82 | 27.65 |
| Ours (F) | 33.82 | 25.25 | 31.28 | 35.43 | 34.40 | 26.83 | 32.37 | 28.13 | 30.94 |
| Ours (HQ) | **34.46** | **25.42** | **31.83** | **36.45** | **35.40** | **27.38** | **33.46** | **28.77** | **31.65** |
| Ours (F-m) | 33.68 | 24.98 | 30.23 | 35.10 | 33.39 | 26.61 | 32.21 | 27.54 | 30.47 |
| Ours (HQ-m) | 34.37 | 25.17 | 30.64 | 36.35 | 34.28 | 27.22 | 33.35 | 28.12 | 31.19 |
| SSIM | | | | | | | | | |
| MobileNeRF | 0.978 | 0.927 | 0.965 | 0.973 | 0.975 | 0.913 | 0.979 | 0.867 | 0.947 |
| nvdiffrec | 0.965 | 0.921 | 0.969 | 0.973 | 0.952 | 0.924 | 0.975 | 0.827 | 0.938 |
| TensoRF (DT) | 0.922 | 0.872 | 0.933 | 0.916 | 0.893 | 0.835 | 0.936 | 0.780 | 0.886 |
| NeuS (DT) | 0.975 | 0.907 | 0.934 | 0.975 | 0.949 | 0.921 | 0.981 | 0.840 | 0.935 |
| nerf2mesh | 0.964 | 0.927 | 0.967 | 0.970 | 0.957 | 0.896 | 0.974 | 0.865 | 0.940 |
| nvdiffrec (m) | 0.970 | 0.915 | 0.937 | 0.973 | 0.943 | 0.927 | 0.975 | 0.820 | 0.932 |
| Ours (F) | 0.977 | 0.935 | 0.974 | 0.978 | 0.978 | 0.925 | 0.981 | 0.865 | 0.952 |
| Ours (HQ) | **0.981** | **0.939** | **0.977** | **0.981** | **0.982** | **0.930** | **0.986** | **0.877** | **0.956** |
| Ours (F-m) | 0.976 | 0.932 | 0.970 | 0.978 | 0.976 | 0.923 | 0.980 | 0.859 | 0.949 |
| Ours (HQ-m) | **0.981** | 0.935 | 0.973 | **0.981** | 0.979 | 0.928 | 0.985 | 0.871 | 0.954 |
| LPIPS | | | | | | | | | |
| MobileNeRF | 0.025 | 0.077 | 0.048 | 0.050 | 0.025 | 0.092 | 0.032 | 0.145 | 0.062 |
| nvdiffrec | 0.023 | 0.086 | 0.032 | 0.064 | 0.047 | 0.111 | 0.031 | 0.188 | 0.073 |
| TensoRF (DT) | 0.076 | 0.130 | 0.070 | 0.113 | 0.090 | 0.146 | 0.070 | 0.230 | 0.115 |
| NeuS (DT) | 0.033 | 0.101 | 0.065 | 0.041 | 0.056 | 0.084 | 0.021 | 0.191 | 0.074 |
| nerf2mesh | 0.046 | 0.084 | 0.045 | 0.060 | 0.047 | 0.107 | 0.042 | 0.145 | 0.072 |
| nvdiffrec (m) | 0.020 | 0.104 | 0.057 | 0.068 | 0.059 | 0.116 | 0.028 | 0.220 | 0.084 |
| Ours (F) | 0.036 | 0.073 | 0.035 | 0.041 | 0.027 | 0.089 | 0.024 | 0.167 | 0.061 |
| Ours (HQ) | **0.026** | **0.068** | **0.033** | **0.035** | **0.023** | **0.085** | **0.017** | **0.159** | **0.056** |
| Ours (F-m) | 0.037 | 0.079 | 0.040 | 0.043 | 0.031 | 0.091 | 0.024 | 0.174 | 0.065 |
| Ours (HQ-m) | 0.027 | 0.074 | 0.038 | 0.036 | 0.027 | 0.086 | **0.017** | 0.164 | 0.059 |

Table 9. Quantitative results on each scene in the NeRF-Synthetic dataset.

| Name | High-Quality | Fast |
|---|---|---|
| app matrix xy | Param (48 x 300 x 300) | Param (48 x 300 x 300) |
| app matrix yz | Param (48 x 300 x 300) | Param (48 x 300 x 300) |
| app matrix zx | Param (48 x 300 x 300) | Param (48 x 300 x 300) |
| app vector x | Param (48 x 300 x 1) | Param (48 x 300 x 1) |
| app vector y | Param (48 x 300 x 1) | Param (48 x 300 x 1) |
| app vector z | Param (48 x 300 x 1) | Param (48 x 300 x 1) |
| basis mat | Linear (144, 12, bias=False) | Linear (144, 27, bias=False) |
| last_layer | Linear (99, 64, bias=True) ReLU (inlace=True) Linear (64, 64, bias=True) ReLU (inlace=True) Linear (64, 3, bias=True) | Spherical Harmonics |

Table 10. Appearance network architecture of Ours (HQ) and Ours (F) for NeRF-Synthetic.

**High-quality.** We use the Vector-Matrix (VM) decomposition in TensoRF, which factorizes a tensor into multiple vectors and matrices along the axes as in Equation 3 of the TensoRF paper. The feature $\mathcal{G}_c(\mathbf{x})$ generated by VM decomposition is concatenated with the viewing direction $d$ and put into the MLP decoder $S$ for the output color $c$:

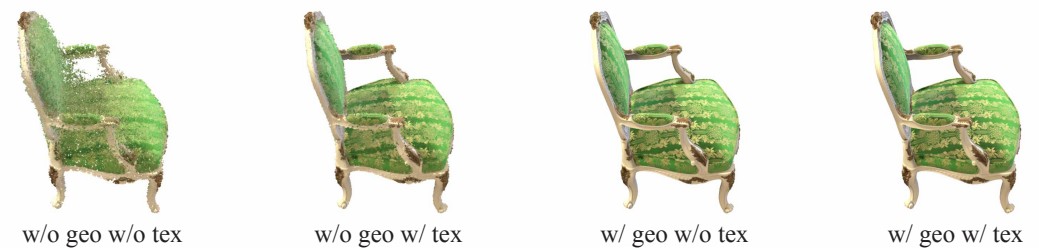

| w/o geo w/o tex | w/o geo w/ tex | w/ geo w/o tex | w/ geo w/ tex |

Fig. 17. Visual comparison of Ours (HQ-m) w/ or w/o geometry and texture initialization. when both initializations are omitted, the mesh optimization process can easily become trapped in local minima, as illustrated in the first left image. Although texture initialization can provide some assistance to the optimization process, it still falls short of achieving satisfactory geometric quality.

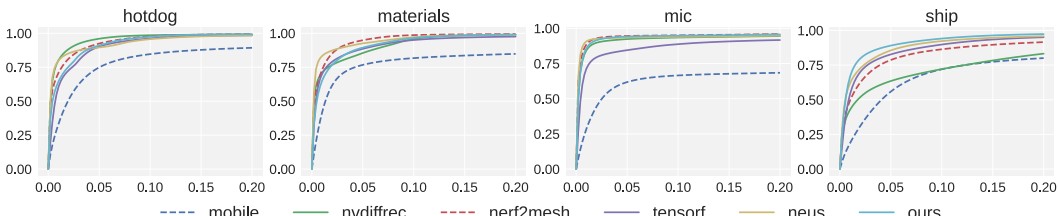

Fig. 18. Other VSA plots.

$$c = S(\mathcal{G}_c(\mathbf{x}), d), \tag{3}$$

We also apply frequency encodings (with Sin and Cos functions) on both the features $\mathcal{G}_c(\mathbf{x})$ and the viewing direction $d$. We use a $300^3$ dense grid to represent the scenes in NeRF-Synthetic and use 2 frequencies for features and 6 frequencies for the viewing direction. The detailed network architecture is shown in Table 10. As for Mip-NeRF 360 and LLFF datasets we use a $512^3$ dense grid to represent the unbounded indoor scenes and do not use frequency encodings.

**Fast.** The fast version shares similar architecture and positional encoding setups with the high-quality version before the MLP decoder but uses the spherical harmonics (SH) function as $\mathcal{G}_c$ instead, as shown in Table 10.

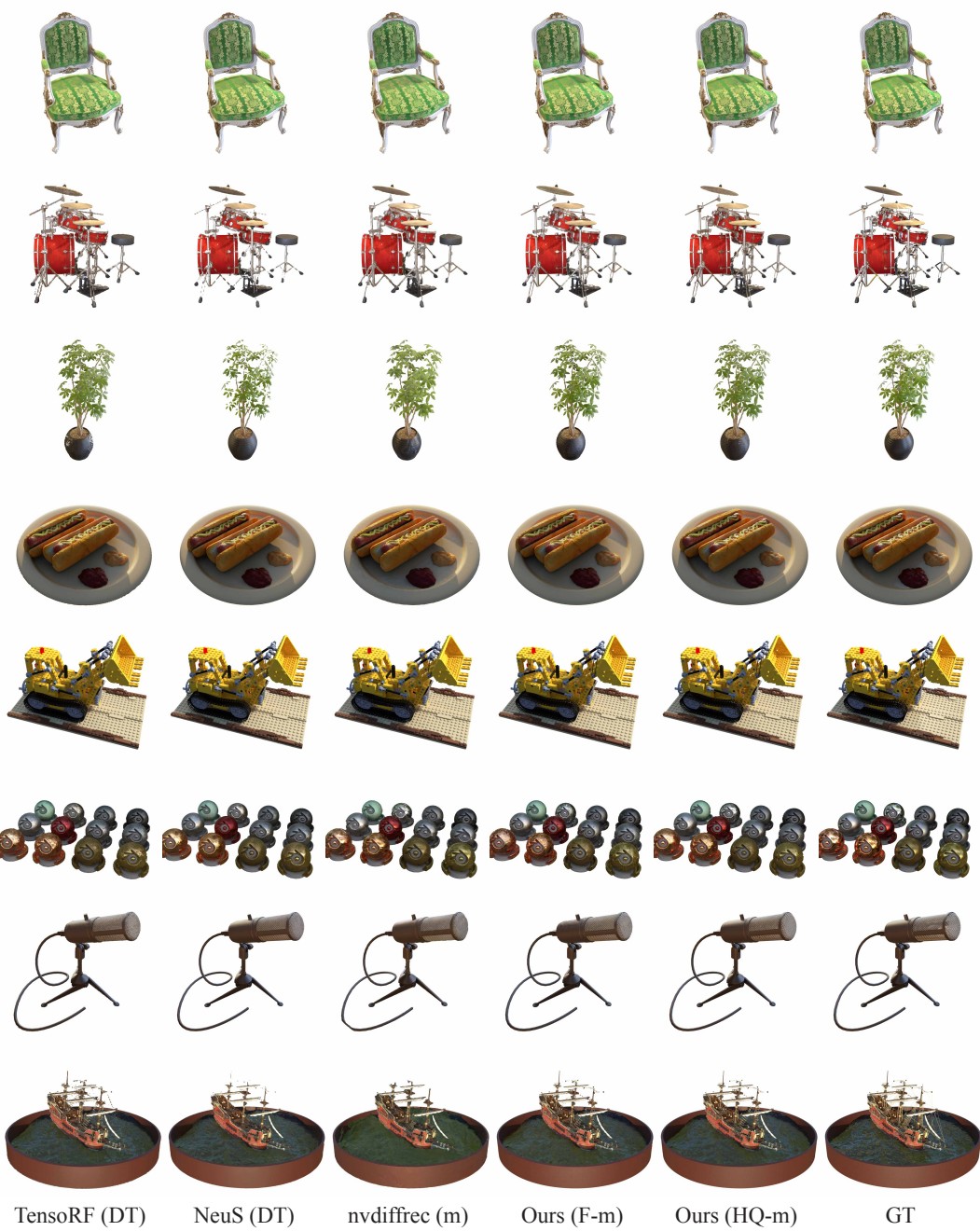

| TensoRF (DT) | NeuS (DT) | nvdiffrec (m) | Ours (F-m) | Ours (HQ-m) | GT |

Fig. 19. NeRF-Synthetic renderings.

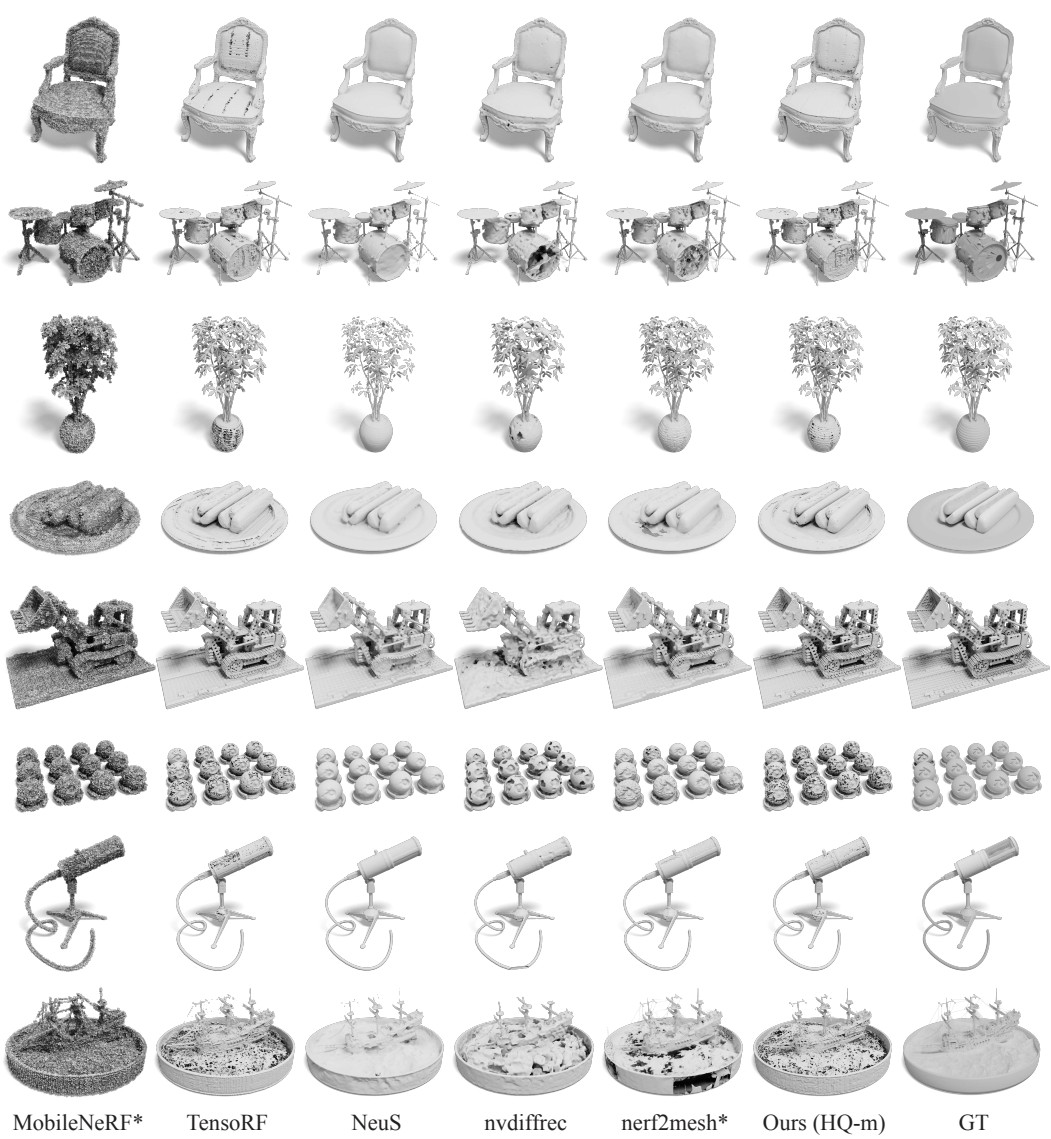

MobileNeRF*    TensoRF    NeuS    nvdiffrec    nerf2mesh*    Ours (HQ-m)    GT

Fig. 20. NeRF-Synthetic mesh.

