# OpenReview forum: "NeuManifold: Neural Watertight Manifold Reconstruction with Efficient and High-Quality Rendering Support"
_ICLR.cc/2024/Conference — Submitted to ICLR 2024_

### Official Review · Reviewer_Q7gG · 2023-10-23

**Soundness:** 3 good
**Presentation:** 3 good
**Contribution:** 2 fair
**Rating:** 3
**Confidence:** 5

**Summary:**

This paper presents a method that is able to generate high quality watertight manifold meshes from multi-view input images. The proposed method starts from a geometry initialized from neural volumetric fields and further optimizes the geometry and a texture representation with differentiable rasterizers. Experiments show that the generated mesh and the texture reconstruction are compatible with existing graphics pipelines.

**Strengths:**

1. The proposed method generates meshes that are applicable to simulation and geometry processing tasks.

2. The proposed method can be integrated with shaders.

**Weaknesses:**

1. The proposed method shown in Figure 2 is a combination of existing papers. For example, volumetric rendering initialization, the geo & app networks and nvdiffrast are all from existing papers. Hence, the novelty of this paper is very limited.

2. How is DiffMC compared to Deep Marching Cubes (https://github.com/yiyiliao/deep_marching_cubes), Neural Marching Cubes (https://github.com/czq142857/NMC), Flexicubes (https://research.nvidia.com/labs/toronto-ai/flexicubes/)? What is the advantage of DiffMC over these methods?

3. While the method is able to generate watertight manifold meshes, it means that the images depicting a scene have some constraints. For example, if the images were depicting the scene of Figure 7 (c) where a blanket is draped onto a chair. I am curious about how the mesh the proposed method reconstructs looks like. The blanket should just be a flat sheet and it shouldn't be represented as a watertight mesh. Similarly, the leaves in Figure 7 (d) shouldn't be represented with watertight meshes.

4. Is the method able to reconstruct objects of a scene into multiple mesh components? For example, in Figure 7 (b) the scene can at least be decomposed into 4 components, a plate, a hotdog and the sauces. Are we able to get four meshes by applying the proposed method to this scene? There are some works looking at understanding the semantics of the scene and perhaps that's good information that can be leveraged to do so.

5. For some physics simulation tasks knowing the internal structure of the object is also important. Is the proposed method infer the internal structure of the reconstructed mesh?

**Questions:**

See the questions above.

---

### Official Review · Reviewer_iTas · 2023-10-29

**Soundness:** 3 good
**Presentation:** 3 good
**Contribution:** 1 poor
**Rating:** 3
**Confidence:** 4

**Summary:**

The manuscript presents a differentiable pipeline for generating watertight manifold meshes from multi-view images observation. The NeuManifold leverages TenorRF [Chen et al, 2022a] to recover a density-based representation. The density is then converted to SDF and later meshes using differentiable marching cubes presented in FlexiCubes [Shen et al, 2023]. The final optimization is baed on nvdiffrast [Laine et al, 2020] is done for improving details. The performance of NeuManifold is compared with prior work on the public available NeRF-Synthetic dataset.

**Strengths:**

-	The paper is well-written and clear to follow. The figures are illustrative to showcase the pipelines and performance variations.
-	The present idea is intuitively sound (except for the first component), with each stage performing meaningful tasks.
-	The problem is of interest to the research community, as image-based watertight reconstruction can be widely applicable in many industries.

**Weaknesses:**

-	Theoretical Limitations
  - The suggested approach is asserted to possess the capability to restore watertight manifold meshes. Nevertheless, these meshes are directly generated from density data with marching cubes, and as a result, there could be arbitrary noisy surfaces extracted, unlike more principled representations such as SDF. The absence of a theoretical framework to ensure a principle density formulation towards watertight mesh contradicts the claimed contributions in the manuscript.

-	Lack of Innovation:
  - Although the overall workflow appears logically coherent, the suggested method primarily amalgamates various pre-existing approaches. The rendering networks draw inspiration from TensorRF [Chen et al, 2022a], the notion of differentiable marching cubes originates from FlexiCubes [Shen et al, 2023], and the concept of differentiable rasterization is credited to nvdiffrast [Laine et al, 2020].


-	Inadequate Experiment Results:
  - The manuscript asserts that "SDF-based techniques often suffer from reduced visual fidelity and the loss of high-frequency geometric details" (Page 4, Section 3.3). However, the paper overlooks recent breakthroughs in SDF-based neural surface reconstruction algorithms capable of generating watertight, high-fidelity meshes from images. Numerous recent advancements, exemplified by Neuralangelo [Li et al, 2023], NeuS2 [Wang et al, 2023], and HF-NeuS [Wang et al, 2022], have emerged in this field. Without comparing against such approaches, it is difficult to understand how much progress NeuManifold has made. Furthermore, building NeuManifold on existing SDF-based approaches seems to be a more logical choice.
  - Moreover, the current results are only constrained to synthetic scenes using NeRF-Synthetic. It will be interesting to see results from real-world captures.

References:

Li, Zhaoshuo, Thomas Müller, Alex Evans, Russell H. Taylor, Mathias Unberath, Ming-Yu Liu, and Chen-Hsuan Lin. "Neuralangelo: High-Fidelity Neural Surface Reconstruction." In Proceedings of the IEEE/CVF Conference on Computer Vision and Pattern Recognition, pp. 8456-8465. 2023.

Wang, Yiming, Qin Han, Marc Habermann, Kostas Daniilidis, Christian Theobalt, and Lingjie Liu. "Neus2: Fast learning of neural implicit surfaces for multi-view reconstruction." In Proceedings of the IEEE/CVF International Conference on Computer Vision, pp. 3295-3306. 2023.

Wang, Yiqun, Ivan Skorokhodov, and Peter Wonka. "Hf-neus: Improved surface reconstruction using high-frequency details." Advances in Neural Information Processing Systems 35 (2022): 1966-1978.

**Questions:**

-	What are the theoretical novelties of the framework such that it is more suitable to the ICLR audience?
- Can authors provide comparisons against the prior works using SDF-based surface reconstruction approaches, such as those listed above?
-	How well does the pipeline work if the NeRF is based on recent SDF-based surface reconstruction methods?
-  How well does the approach work for real-world captures?
-	The manuscript claims in Figure 5 that it is safe to assume axis alignment between marching cubes coordinates and real-world objects. Without explicit alignment, this may not be the case. Can the authors provide more details on this?
-	Figure 3 compares marching cubes/tetrahedral results with and without “exp()” on SDF fields. Why is this important given SDF is already meaningful for surface extraction?

---

### Official Review · Reviewer_1eXC · 2023-10-31

**Soundness:** 2 fair
**Presentation:** 3 good
**Contribution:** 2 fair
**Rating:** 3
**Confidence:** 4

**Summary:**

This paper proposes a geometry reconstruction method, based on the following: neural radiance fields, a differentiable form of marching cubes, and differentiable rasterization.  They leverage components from existing works such as TensoRF and nvdiffrast, in addition to implementing the differentiable MC part in CUDA for high performance.  The pipeline results in a high-quality, watertight manifold mesh, along with a "neural texture", that can be used for rendering and simulation.

**Strengths:**

The writing quality is generally good, and well-articulated.  The paper provides thorough evaluation on multiple datasets, and comparisons to multiple baseline methods.  The end-to-end differentiability of the system allows refinement of the initial geometry based on a rendering loss, which improves the quality of the reconstruction result.  The implementation and engineering, such as deploying the optimized model in GLSL shaders, is notable.

**Weaknesses:**

While the paper shows favorable results in terms of quality of reconstruction, it seems only about on par or marginally better than methods that it has been compared to.  It mostly connects components from existing works (TensoRF, nvdiffrast, Marching Cubes algorithm) into a well-engineered end-to-end pipeline, but I question whether the specific technical contributions of the paper are significant enough to stand on their own.

The paper claims that it is the first differentiable Marching Cubes implementation, yet it's not clear whether this is indeed the case (see for example: Deep Marching Cubes, MeshSDF, Neural Marching Cubes, Flexicubes).  Furthermore, the differentiable formulation of MC presented in the paper is compared only to DMTet, but not to any other methods that might be even closer in spirit (i.e. those mentioned in the previous remark).  Therefore, it is hard to evaluate the level of contribution.

**Questions:**

- The paper strongly emphasizes the watertight manifoldness of the resulting meshes, but does not go into much detail about the specific technical contribution of this paper that results in this property.  Could you clarify whether this is indeed a contribution of this paper, or is it inherited from the other methods that the paper builds off of?

- On page 4, it is stated that during the second, mesh optimization step, the *topology* is also optimized.  Could you elaborate/clarify what you mean by this exactly?

- On page 5: "given that most real-world objects tend to be axis-aligned" -- could you briefly explain why this is a reasonable assertion to make?

- SDF-based reconstruction methods (e.g. NeuS) claim that the raw density field is optimized only to satisfy the rendering objective (and is also biased away from the true surface), and therefore are not ideal for surface reconstruction.  In this paper, you argue that this is unnecessary and that using the density field directly is a better choice.  Could you elaborate on this choice?

---

### Official Review · Reviewer_XSZW · 2023-11-01

**Soundness:** 2 fair
**Presentation:** 3 good
**Contribution:** 2 fair
**Rating:** 5
**Confidence:** 4

**Summary:**

This paper introduces NeuManifold, a method for creating watertight manifold meshes from multi-view images. The approach integrates neural field rendering with differentiable rasterization-based mesh reconstruction techniques, providing accurate reconstructions with authentic appearance. The contribution lies in introducing Differentiable Marching Cubes (DiffMC) to yield smoother surfaces, coupled with seamless integration with GLSL shaders for real-time rendering.

**Strengths:**

Originality: This method is a synthesis of various existing technologies. TensoRF provides initialization for the subsequent mesh optimization. Differentiable Marching Cubes further refine the mesh, and nvdiffrast is used for rendering. The entire process is amenable to gradient backpropagation.

Quality: Experiments indicate an improvement in both appearance synthesis and mesh generation quality compared to previous research and existing methods.

Clarity: The language used for explanation is generally concise.

Significance: The differentiable Marching Cubes hold significance and can serve as inspiration for other related fields.

**Weaknesses:**

1. The majority of this method is a combination of previous works. DiffMC demonstrates a certain level of innovation, but the explanation of this part of the method is not sufficiently clear. For example, the specific implementation of the approach in this paper and its differences from the "Deep marching cubes: Learning explicit surface representations" paper are not discussed.

2. The main text claims a 10x speedup in DiffMC compared to DMTet. However, this improvement is not intuitively and reasonably explained. Additionally, the speed comparison between DiffMC and DMTet should be reflected in the experimental section.

3. The paper claims that the method can generate watertight and manifold meshes, but marching cubes do not necessarily guarantee manifold properties in regions with complex iso-surfaces. The authors should compare whether the proportion of manifold vertices generated in different scenarios has improved. Otherwise, this claim might be somewhat exaggerated.

4. The specific details of how the finetuning is performed in the third stage are not well explained.

**Questions:**

1. What are the specific working details of DiffMC? Please provide more explanation.

2. Can the proposed method completely guarantee the watertightness and manifold of the generated mesh?

3. Why is the proposed method faster than DMTet? Please provide more explanation.

---

### Meta-Review · Area_Chair_eSc8 · 2023-12-07

**Metareview:**

This work aimed at recovering 3D from multi-view images and proposed to combine mesh representations with volumetric optimization via differentiable marching cubes. The reviewers raised several concerns, in particular regarding the strength of the contributions (as this work primarily combined components from prior) and the benefits of DiffMC in context of DMTet and prior differentiable marching cubes implementations. The authors did not respond to the concerns raised, the reviewers unanimously recommend rejection. The AC concurs with this recommendation.

**Justification For Why Not Higher Score:**

There are several concern regarding contributions and lack of comparisons to prior differentiable marching cube implementations. While these are unresolved, the paper cannot be recommended for acceptance.

**Justification For Why Not Lower Score:**

N/A

---

### Decision · Program_Chairs · 2024-01-16

Reject